# ASIA Syndrome: State-of-the-Art and Future Perspectives

**DOI:** 10.3390/vaccines12101183

**Published:** 2024-10-17

**Authors:** Mario Caldarelli, Pierluigi Rio, Vincenzo Giambra, Antonio Gasbarrini, Giovanni Gambassi, Rossella Cianci

**Affiliations:** 1Department of Translational Medicine and Surgery, Catholic University of Sacred Heart, 00168 Rome, Italy; mario.caldarelli01@icatt.it (M.C.); pierluigi.rio01@icatt.it (P.R.); antonio.gasbarrini@unicatt.it (A.G.); giovanni.gambassi@unicatt.it (G.G.); 2Fondazione Policlinico Universitario A. Gemelli, Istituto di Ricerca e Cura a Carattere Scientifico (IRCCS), 00168 Rome, Italy; 3Institute for Stem Cell Biology, Regenerative Medicine and Innovative Therapies (ISBReMIT), Fondazione IRCCS “Casa Sollievo della Sofferenza”, 71013 San Giovanni Rotondo, Italy; v.giambra@operapadrepio.it

**Keywords:** ASIA, genetic predisposition, adjuvants, vaccines, immune system

## Abstract

The expression “Autoimmune/inflammatory syndrome induced by adjuvants (ASIA)” was coined by Shoenfeld and colleagues in 2011. It defines a group of immune-mediated disorders that arise in people, with a genetic predisposition, following exposure to adjuvant agents. This syndrome has been reported after contact with silicone implants, medications, infections, metals, vaccines, and other substances. It typically occurs in individuals with a genetic predisposition, particularly involving genes, such as *HLA-DRB1* (major histocompatibility complex, class II, DR beta 1) and *PTPN22* (protein tyrosine phosphatase non-receptor type 22). Some stimuli lead to an overactivation of the immune system, prompt the production of autoantibodies, and finally cause autoimmune disorders. This narrative review aims to provide an overview of the ASIA syndrome with a special focus on the role of adjuvants in different vaccines, especially after the COVID-19 pandemic, and insights into development of new treatments.

## 1. Introduction

The Autoimmune/inflammatory Syndrome Induced by Adjuvants (ASIA), was first coined by Shoenfeld et al. in 2011 [1].

The term ASIA encompasses a range of clinical manifestations, associated with exposure to various adjuvants, that have in common the production of non-specific autoantibodies and a breakdown of immune tolerance [2].

This narrative literature review aims to elucidate the pathogenesis of ASIA syndrome, differentiate it from other established autoimmune diseases, and outline potential avenues for future research in this area.

In addition, a deeper understanding of ASIA could lead to the development of safer vaccine adjuvants. Investigation on the pathogenesis of ASIA supports the use of novel biologic agents for treatment, such as IL-1 receptor antagonists or JAK inhibitors [3,4].

ASIA includes various medical conditions: vaccination-induced autoimmune disorders, siliconosis, Gulf War Syndrome (GWS), macrophagic myofasciitis (MMF) with chronic fatigue syndrome, and sick-building syndrome (SBS) [5].

To date, GWS is a condition only partially understood [6]. Symptoms such as muscle weakness, myalgia, fatigue, ataxia, cognitive dysfunction, sweating disorder, fever, arthralgia, headache, skin rash, gastrointestinal and bladder disorders, and sleep disturbances have been reported.

Additionally, individuals with GWS may experience higher chemical sensitivity and intolerance to odors. The exact cause of GWS is not well-defined; however, links to pyridostigmine bromide, specific vaccination regimens, various chemical exposures (e.g., smoke from oil-well fires or uranium from munitions), physical and psychological stress have been reported in the literature [3].

MMF is an inflammatory histopathological condition characterized by the presence of aluminum adjuvants, derived from vaccines, within muscle phagocytes. Patients with MMF commonly experience generalized myalgia and fatigue, symptoms that align with myalgic encephalomyelitis/chronic fatigue syndrome (ME/CFS) [7]. MMF phagocytes appear to utilize microtubule-associated proteins 1A/1B light chain 3′-associated phagocytosis (LAP) to process aluminum oxyhydroxide vaccine particles and show an intensified metabolic response to vaccines [8].

SBS is a collection of skin, mucosal, and systemic symptoms linked to time spent in residential buildings, though the exact causes are unclear [9]. It is significantly associated with the absence of functional windows, recent use of pesticides, tints, and diluents, indoor cooking with polluting fuels, fungal growth in the buildings, air pollution, and house dust.

These diseases may be related to ASIA, as they are influenced by various environmental factors with immune-adjuvant properties, such as silicone and pollutants, which have been implicated in the development of both defined and undefined immune-mediated diseases [10].

The presence of an adjuvant that constantly stimulates the immune system is the key element of ASIA. It is a component typically included in vaccines to enhance and boost the immune response to a specific antigen, leading to higher antibody production against pathogens. Other substances that may also trigger ASIA, though not used in vaccines, are silicon and heavy metals, such as mineral oil, mercury, and titanium [11].

Since the definition of ASIA in 2011, up until 2016, over 4000 patients have been diagnosed. Many cases have been reported to be linked to human papillomavirus (HPV) and influenza vaccines and with mineral oil fillers and silicone implants [12].

Similarly, during the Coronavirus 2019 (COVID-19) pandemic, ASIA-like symptoms were reported in people vaccinated with the COVID-19 vaccine [13].

ASIA summarized and categorized vaccine-related adverse events under a single term, a step that highlighted the need for improved immune stimulants beyond traditional adjuvants [14]. The COVID-19 pandemic highlighted new technologies and vaccine development methods, particularly with the introduction of mRNA-based vaccines [14].

This narrative review was conducted by searching electronic databases, such as PubMed, MEDLINE, and Google Scholar, using keywords such as ‘ASIA’, ‘adjuvant-induced autoimmune and inflammatory syndrome’, ‘inflammation’, and ‘adjuvant’. We considered original and review articles, written in English, published in peer-reviewed journals since 2011. The articles have been selected based on topic, study design, methodology, and sample size, including both small- and large-sample studies, and case report, since ASIA is a relative recently described syndrome. To increase the number of studies to include in this review, a manual search of cited articles was also conducted.

## 2. Epidemiology

Since 2011, more than 4479 cases of ASIA syndrome have been reported. Approximately 92.7% of these occurred in women and were mainly associated with adjuvants in HPV and hepatitis B (HBV) vaccines. In a systematic review by Jara et al., severe cases accounted for 6.8% of the total, with a mortality rate of 0.24% [15].

The prevalence varies widely, from 0.5% to 25.7%, with clinical manifestations often resembling polygenic autoimmune diseases, such as rheumatoid arthritis and systemic lupus erythematosus. Autoinflammatory conditions are less common, occurring in 0.5% to 2.5% of cases.

Hennekens et al. conducted a retrospective cohort study comparing 10,830 women with silicone breast implants to those without, assessing the risk of connective tissue diseases linked to ASIA [16].

The association between ASIA syndrome and implant exposure remains controversial. Studies suggest that an association often has a high risk of bias, mainly due to reliance on self-reported symptoms and inadequate long-term follow-up. The occurrence of autoimmune diseases appears to be rare and is unlikely to be directly related to implants [17].

In addition, Rohan et al. reported that patients receiving allergen-specific immunotherapy are exposed 100 to 500 times more to aluminum than patients receiving hepatitis B or human papillomavirus vaccines [18].

Studies have shown that patients receiving allergen immunotherapy with aluminum-containing products have a lower risk of autoimmune disease compared with reported cases of ASIA. In addition, a clinical trial found no increase in symptoms in patients with systemic lupus erythematosus who received the hepatitis B vaccine [19].

Thus, several researchers argue that current evidence does not establish a causal relationship between aluminum-containing vaccine adjuvants and ASIA. This should be reassuring individuals receiving routine vaccinations or allergen-specific immunotherapy.

Alternative study designs that rely on vaccine company data or voluntary government reporting may be prone to bias, particularly underreporting [18]. Given the widespread use of aqueous and glycerinated allergen products in the United States, future research could compare autoimmune rates between American and European patients undergoing allergen immunotherapy.

## 3. Clinical Presentation

According to Shoenfeld et al., the diagnosis of ASIA requires two major criteria, or one major and two minor criteria, as outlined in Table 1 [20].

Watad et al. analyzed data from the ASIA syndrome registry since 2011 to February 2019, examining 500 cases of the syndrome. The most common symptoms were joint and muscle pain, and chronic fatigue, with a mean latency period varying from 3 days to 5 years after the administration of the vaccine [20].

In a systematic review, Shoenfeld et al. listed arthralgia, arthritis, chronic fatigue, myalgia, sleep disturbances, general malaise, mucosal dryness, fever, and neurological symptoms as the most commonly reported symptoms [1].

However, ASIA is a comprehensive term that covers various conditions, like sarcoidosis, Sjögren syndrome (SS), undifferentiated connective tissue disease (UCTD), and silicone implant incompatibility syndrome that share similar pathophysiology.

For example, women who experience silicone-related problems with silicone breast implants (SBIs) have been included in traditional models of ASIA syndrome.

Silicone in breast implants can chronically stimulate the immune system in genetically predisposed individuals, leading to non-specific subjective clinical outcomes and potentially causing autoimmune diseases and lymphomas [21].

It has been observed that mice develop proteinuria and autoimmune hemolytic anemia following the injection of silicone gel. Moreover, the implantation of silicone gel or silicone oil in homozygous mice for the gene *LPR* (a gene that induces marked lymphoproliferation) has increased anti-double-stranded (ds) DNA antibodies [22].

Recently, Borba et al. diagnosed 100 women with SBIs who were experiencing classical ASIA-related clinical manifestations, such as chronic fatigue, sleep disturbances, generalized pain, dry mouth and eyes, cognitive decline, palpitations, hearing abnormalities, allergic reactions, mood disorders, hair loss, irritable bladder and bowel syndrome [21].

Moreover, the same study group reported elevated secretion of a wide variety of autoantibodies in women with SBIs [23].

Recently, an increasing number of people have been complaining of chronic fatigue, muscle pain, muscle weakness, joint pain, arthritis, and interstitial lung disease. This symptomatology has been categorized as UCTD, which includes a range of symptoms, signs, and laboratory findings indicative of systemic autoimmune diseases [24].

Scanzi et al. reported that exposure to multiple adjuvants before the onset of UCTD was significantly higher in patients compared to healthy controls, suggesting that approximately half of UCTD patients may fall on the ASIA spectrum [25].

Another situation that may be reclassified to the ASIA group is immune-related adverse events caused by checkpoint inhibitors in cancer therapy. In this case, the triggering particles are monoclonal antibodies that inhibit receptors expressed by T cells, such as cytotoxic T lymphocyte-associated antigen-4 (CTLA-4), programmed cell death protein 1 (PD-1), and its ligand PD-L1. Inhibition of these checkpoint receptors weakens the inhibition of self-recognition by lymphocytes, leading to activation of CD8+ and CD4+ T cells against cancer cells. This can overstimulate the immune system and lead to autoimmune reactions [26].

It has been found that 2–12% of cases developed the typical ASIA symptoms, like arthritis, myositis, and conditions resembling polymyalgia [27].

Interestingly, ASIA is often associated with autoimmune endocrine disorders. In the literature, 52 events of sub-acute autoimmune thyroiditis following contact with adjuvants have been described. These include 41 cases after the HPV vaccine, 8 events following the influenza vaccination, 1 case after exposure to the HBV vaccine, 1 following a silicone breast implant, and 1 following exposure to mineral oils [28].

Vayssairat et al. reported two cases of autoimmune Hashimoto’s thyroiditis after the receipt of silicone gel-filled breast implants [29]. Patients were positive for antinuclear antibodies (ANA) and anti-thyroid microsomal autoantibodies [29].

Mochizuki et al. have recently described two cases of type 1 diabetes mellitus following the administration of multiple doses of severe acute respiratory syndrome coronavirus 2 (SARS-CoV-2) vaccines [30]. Similarly, Aydoğan and colleagues reported four cases of vaccine-induced autoimmune diabetes after BNT162b2 (Pfizer-BioNTech) vaccination against SARS-CoV-2. All patients were positive for anti-glutamic acid decarboxylase (GAD) autoantibodies [31].

## 4. Pathophysiology

ASIA is a complex syndrome determined by a synergy of both genetic and environmental elements [32].

The development of the ASIA syndrome is based on the idea that early exposure to an adjuvant can trigger a series of biological and immunological processes which, in susceptible individuals, may eventually lead to the onset of autoimmune diseases [33].

Genetic factors may contribute to the development of the syndrome, although they are minor diagnostic criteria.

Given the significant exposure to environmental factors and the relatively low prevalence of ASIA, the involvement of epigenetic mechanisms was hypothesized [32].

The genetic relationship is influenced by specific human leukocyte antigen (HLA) antigens involved in autoimmune diseases [34]. This syndrome is commonly related with the presence of HLA-DRB1, and HLA-B27 [21].

Individuals with HLA-B27 are more likely to develop autoimmune diseases such as uveitis, Reiter’s syndrome, and ankylosing spondylitis after vaccination, supporting the “mosaic of autoimmunity” theory [14]. Another gene associated with this syndrome is protein tyrosine phosphatase non-receptor type 22 (PTPN22), which encodes a member of the protein tyrosine phosphatase family. The PTPN22 gene plays a critical role in the adaptive immune system by regulating both T and B cells. Variants of this gene, particularly single nucleotide polymorphisms (SNPs), have been shown to significantly impair various immune functions, leading to dysregulation of immune responses [35].

Mutations in this gene are found in various autoimmune diseases [21].

ASIA appears to have a higher prevalence in women. Watad et al. analyzed a registry of 500 ASIA cases, and they found that 89% of the cases were women [20].

In animals, exposure to adjuvants results in signs and symptoms like those seen in human ASIA. Lujan and colleagues found that ASIA developed in commercial sheep following exposure to adjuvants in blue tongue vaccine [36].

Most of the models studied in the laboratory to describe the natural history of ASIA use aluminum as a reference adjuvant.

In 2002, HogenEsch et al. studied how aluminum compounds act as adjuvants. They found that aluminum salts trigger the activation of dendritic cells (DCs) and complement components, and increase the secretion of chemokines at the injection site [37].

Recently, activation of the nucleotide-binding domain, leucine-rich–containing family, pyrin domain–containing-3 (NLRP3) inflammasome has been implicated in the adjuvant effect of aluminum [38].

The inflammasome is an intracellular multiprotein complex that facilitates the cleavage of the inactive precursor of the proinflammatory cytokine interleukin (IL)-1β by caspase-1, leading to the release of mature IL-1β [39]. Inflammasome-mediated cleavage of pro-IL-1β in vitro relies on signals that activate both Toll-like receptors (TLRs) and NOD-like receptors (NLRs), such as NLRP3. Activation of these innate immune receptors is now understood to be essential for effective adaptive immunity, providing the necessary combination of stimuli to naïve T cells [38].

The aluminum particles cause damage to the membrane of the lysosome, which in turn activates NLRP3 through the action of cathepsin B [40].

Flach et al. have shown in vivo that in the presence of an inflammasome deficiency, several pathways that can trigger the immune response are activated [41]. They report that alum (aluminum hydroxide and other aluminum salts) does not have a receptor on the surface of DCs. Instead, it interacts directly with lipids in the plasma membrane of DCs, causing lipid sorting like the effects of monosodium urate crystals. This interaction leads to the aggregation of immunoreceptor signaling motif (ITAM)—containing receptors and triggers phagocytic responses mediated by spleen tyrosine kinase (Syk) and phosphoinositide 3-kinase (PI3K). Alum does not enter the cell but facilitates the transport of the mixed soluble antigen across the plasma membrane. As a result, DCs exposed to alum develop a strong affinity for CD4+ T cells, facilitating T helper 2 (Th2) immune response and the subsequent activation of B cells. The evidence presented suggests that this strong binding is mediated by intercellular adhesion molecule-1 (ICAM-1) and lymphocyte function-associated antigen-1 (LFA-1) [41]. They also showed that IL-1β production was absent in both NLRP3−/− DCs in response to alum, while the expression levels of tumor necrosis factor-α (TNF-α) and the co-stimulatory molecules CD80, CD86, and CD40 were like those in wild-type DCs. These results suggest that membrane events unrelated to the NLRP3 inflammasome are sufficient to mediate DCs activation in response to alum [41].

Danielsson and Eriksson highlighted that the intracellular accumulation of aluminum ions is responsible for the polarization of macrophages into inflammatory and antigen presenting M1 macrophages through various mechanisms, including the inhibition of phagosomes pH reduction, the increased production of reactive oxygen species (ROS), the damage of the phagosomal membrane, the extracellular release of damage-associated molecular patterns (DAMPs) and metabolic changes [42].

Furthermore, reactive oxygen species (ROS) and reactive nitrogen species are generated, leading to macrophage apoptosis [43]. This process releases agent particles, such as silica-containing particles, which can be reabsorbed by macrophages. In addition, exposure to these silica particles triggers a significant production of IL-17, leading to the activation of neutrophils, the production of ROS, and the release of enzymes such as myeloperoxidase [43].

Another aspect of adjuvants is their association with uric acid. Crystallized uric acid is recognized as a natural endogenous danger signal [44]. Alum, for example, is thought to trigger an inflammatory response that leads to the release of uric acid from necrotic cells. Uric acid enhances the adjuvanticity of alum, causing an apparent increase in IL-4 levels. Therefore, it is not surprising that inhibiting the formation of uric acid or promoting its degradation results in a reduction in the adjuvanticity of alum, as indicated by a further reduction of IL-4 levels [37].

Another danger signal that can enhance the adjuvanticity of alum is the DNA derived from the host cell, which is released by necrotic cells [45].

Other adjuvants, unlike aluminum, enhance specific adaptive immune responses by targeting innate immune cells and activating pattern-recognition receptor (PRR) signaling pathways. Examples of such adjuvants include those targeting nucleotide-binding oligomerization domain 1 (NOD1), NOD2, NLRP3, retinoic acid-induced gene I (RIG-I), and melanoma differentiation-associated gene 5 (MDA5) [46]. The activation of these receptors by adjuvants leads to the upregulation of major histocompatibility complex (MHC) class II by antigen presenting cells (APC), thereby contributing to enhanced antigen presentation [47].

In addition, activation of NOD1, NOD2, and NLRP3 leads to the production of pro-inflammatory cytokines such as IL-1β and IL-18, which promote the polarization of naïve T cells into Th2-type cells [48]. For example, muramyl dipeptide (MDP) or complete Freund’s adjuvant (CFA) can activate NOD1 and NOD2, leading to stimulation of nuclear factor kappa-light-chain-enhancer of activated B cells (NF-κB) [47]. This results in the production of IL-1, IL-18, and IL-33 precursors. In the presence of the activated NLRP3 inflammasome complex, caspase-1 subsequently cleaves these precursors into their active forms [49].

In contrast, retinoic acid-inducible gene I (RIG-I) and melanoma differentiation-associated protein 5 (MDA5) are members of the retinoic acid-inducible gene I-like receptor (RLR) family, which are primarily responsible for recognizing RNA [50]. Most TLR3 agonists, such as poly-I, also activate MDA5 in APCs [51]. RIG-I and MDA5 trigger the activation of interferon regulatory factor 3 (IRF3) and interferon regulatory factor 7 (IRF7), which subsequently induce the expression of type I interferons. Therefore, immunostimulants targeting these two receptors are biased towards promoting the production of Th1 cells and cytotoxic T lymphocytes (CTLs) [52], and ultimately the activation of the adaptive immune system.

Pathogen-associated molecular patterns (PAMPs) expressed by pathogens can activate PRRs on DCs. This interaction leads to the upregulation of MHC class II and costimulatory molecules on DCs and triggers the secretion of the Th1-polarising cytokine IL-12 [53]. When naïve, CD4+ T cells are activated in the presence of IL-12 or interferon (IFN)-gamma. They upregulate T-bet, which directs these T cells to differentiate into Th1 lineage cells and produce Th1 cytokines such as IFN-gamma [54]. IL-12 can induce the differentiation of T follicular helper (TFH) cells capable of producing IFN-gamma, known as TFH1 cells [55]. Interestingly, B cells can also produce IL-12 and IFN-gamma and, under certain conditions, play a crucial role in the expansion or maintenance of the Th1 response [56]. Given the important role of both DCs and B cells in the development and maintenance of TFH cells, it is likely that B cells and DCs work together to optimize the induction of TFH1 and Th1 responses [57].

A similar cytokine-dependent model was originally proposed for Th2 development, where the lineage-driving cytokine IL-4, produced by APCs in response to pathogen-directed signals, induces the upregulation of the Th2 lineage-specific transcription factor GATA-3 in T cells [58].

Dong et al. have suggested that Th2 development may occur as a default pathway in the absence of Th1-polarising cytokines or PAMPs/DAMPs that can activate PRRs expressed by DCs [59].

B cells can secrete several cytokines, including IL-12 and IFN-gamma, that can promote T-box transcription factor (TBX21) expression in T cells and facilitate Th1 differentiation [60]. In addition, B cells produce IL-10, which can suppress IL-12 production by DCs [61]. In turn, by producing IL-2 and IFN-gamma, Th2 cells promote the maturation of B cells into plasma cells [2].

Some researchers consider adjuvants to be environmental factors that contribute to autoimmune diseases. However, supplementation of apoptotic cells with strong adjuvant signals often fails to induce clinical autoimmunity in most strains; the resulting autoantibodies are typically transient, do not undergo epitope spreading, and do not lead to disease [62].

Adjuvants may also play a dual role in the mechanisms underlying autoinflammatory and autoimmune diseases. Myasthenia gravis (MG) and its animal model, experimental autoimmune myasthenia gravis (EAMG), result from the interference with neuromuscular transmission by autoantibodies targeting the nicotinic acetylcholine receptor (AChR) on muscle cells [63]. Two peptides, referred to as RhCA 67-16 and RhCA 611-001, have been designed to structurally complement the main immunogenic region and the dominant T-cell epitope of the AChR. These peptides serve as effective vaccines that prevent EAMG in rats by inducing anti-idiotypic/clonotypic antibodies and reducing levels of AChR antibodies [63].

McAnally et al. tested the efficacy of RhCA 611-001 in combination with several adjuvants approved for use in humans [63]. The adjuvants chosen for comparison were incomplete Freund’s adjuvant (IFA) and aluminum hydroxide. The study showed that disease protection was qualitatively, but not quantitatively, related to the anti-peptide antibody response.

As observed in various inflammatory and autoimmune diseases, the Janus kinase-transcriptional signal transducer activator (JAK/STAT) pathway may also play a role in pathogenesis.

The JAK/STAT signaling pathway is central to cytokine signaling and plays an important role in development, immune response, and tumorigenesis in almost all cell types [64]. Several factors influence JAK/STAT signaling activity, including cytokine diversity, receptor profiles, overlapping specificities of JAK and STAT proteins, and both positive regulators—such as co-operating transcription factors—and negative regulators—such as Suppressor of Cytokine Signaling, protein inhibitors of activated STAT (PIAS) [65]. These elements highlight the complex structure of the pathway, which is easily disrupted by mutations. The JAK/STAT pathway has been a focus of basic research and continues to hold great promise for the development of new approaches to personalized medicine, advancing the translation of molecular research into clinical applications beyond JAK inhibitors. Mutations that enhance or reduce the function of key signaling molecules such as STAT1, STAT3, STAT6, JAK1, and JAK3 are associated with distinct clinical phenotypes [66]. The traditional view that loss-of-function mutations cause immunodeficiency and gain-of-function mutations lead to autoimmunity is becoming obsolete, giving way to a better understanding of disease patterns [64].

For example, Gruber et al. describe a patient with early-onset multi-organ immune dysregulation caused by a mosaic gain-of-function mutation (S703I) in JAK1, a kinase critical for signaling downstream of more than 25 cytokines [67]. Using custom single-cell RNA sequencing, they investigated the mosaicism at single-cell resolution. They discovered that JAK1 transcription was mainly restricted to one allele in different cells, leading to the concept of a mutant “transcriptotype” that is distinct from the genotype. This patient was treated with tofacitinib, a JAK inhibitor, resulting in rapid clinical resolution.

In addition, Flanagan et al. report a newly identified monogenic cause of autoimmunity due to de novo germline activating mutations in STAT3. This discovery was made in five individuals with a variety of early-onset autoimmune diseases, including type 1 diabetes [68].

As reported by Schwartz et al., cytokine signaling could theoretically be blocked by inhibiting STAT activation, disrupting STAT-receptor interactions, preventing STAT dimerization, or interfering with STAT-DNA binding [69]. These approaches have been considered particularly in cancer, where persistent JAK/STAT activation is common [70]. However, unlike JAKs, STATs are not enzymes, making the development of clinically viable drug candidates challenging due to issues of bioavailability, in vivo efficacy, and selectivity [71]. For example, STAT3, a widely studied target in the treatment of solid organ malignancies, shares a high degree of homology with STAT1.

The pathogenetic mechanisms mentioned above are summarized in Figure 1.

## 5. ASIA and Adjuvants

Adjuvants are vaccine constituents allowing a more effective immune response towards antigens, by increasing the stability, the half-life, and the immunogenicity of the antigen element, and diminishing the number of vaccine doses administered.

In order to achieve this goal, organic and inorganic compounds, mineral oils, macromolecules, detergents and microbial products have been employed [72].

According to their mechanism of action, adjuvants are classified into different groups: delivery systems (e.g., mineral salts, microparticles, and emulsions), immune potentiators (e.g., TLR agonists), mucosal adjuvants and combined adjuvants [73]. The delivery system is an antigen carrier and the vaccine facilitates immune cell recruitment and antigen presentation at the injection site, enhancing both innate and adaptive immune responses. On the other side, immune potentiators are TLR agonists that promote the production of antigen signals and co-stimulatory signals, enhancing adaptive immune responses. Furthermore, IFN induces the release of inflammatory cytokines such as IL-12, TNF-α, and IL-1β and activates inflammatory pathways such as the myeloid differentiation primary response 88 pathway (MYD88) [49,74]. Mucosal vaccine adjuvants include bacterial toxins, nanoparticles, biopolymers, cytokines, and chemokines, able to potentiate the immune responses against antigens at a mucosal level (e.g., improving mucosal barrier, secretions, and cellular immune responses) [75].

### 5.1. Vaccines

Autoimmune or immune-mediated diseases following vaccination, not necessarily meeting the criteria of ASIA, have been identified.

The various adjuvants, their associated vaccines and their mechanisms are summarized in Table 2. Aluminum adjuvants are mostly used in vaccinology due to their safety, availability, and cost-effectiveness. They also serve to stabilize vaccine components and ensure the preparation of suitable vaccine formulations [76]. For this reason, aluminum is a component of several vaccines, including those against tetanus, influenza, pneumococcus, and hepatitis A and B [32].

Aluminum is responsible for the triggering of NLRP3 inflammasome with the subsequent secretion of pro-inflammatory cytokines, such as IL-1β, enhances phagocytosis and reduces the release of antigens from the injection site, thus allowing inflammatory cells to gather [32].

Aluminum adjuvants are well-known enhancer of Th2 response. However, it has recently been suggested that aluminum induces Th1 responses in the presence of other Th1-promoting compounds (e.g., lipopolysaccharide or recombinant influenza antigen). Furthermore, a bystander effect through which aluminum adjuvants trigger autoimmunity has been described, consisting in the activation of dormant autoreactive T lymphocytes in some individuals [77].

Moreover, aluminum adjuvants are known to stimulate a type of immune response associated with increased levels of IgE and eosinophils, which are associated with allergic reactions [78].

Honda et al. found that vaccines containing aluminum adjuvants can induce allergic-type lung inflammation in mice, characterized by increased eosinophils number [79]. 

Replacing aluminum adjuvants with delta inulin-based ones may help prevent this type of reaction [80].

A strong inflammatory response can produce cytokines (IL-4, IL-5, and IL-13) that can damage the lungs and the heart. This inflammation can spread from the lungs to the heart, causing pericarditis, myocarditis, and irregular heartbeats [81].

Gherardi et al. showed that inflammatory lesions linked to aluminum hydroxide were detectable at injection sites for an extended period in patients who experienced widespread muscle pain after receiving aluminum-based vaccines [82].

Horning et al. reported that patients with chronic myalgic encephalomyelitis/chronic fatigue syndrome (ME/CFS) exhibit an exhausted immune system, as evidenced by significant decreases in IL-1b, IL-1ra, IL-4, IL-10, IL-12, IL-17, and increases in chemokine (C-C motif) ligand 2 (CCL2) and the major monocyte chemoattractant [83]. Polyoxyethylene sorbitan monooleate and sorbitan trioleate (MF59) is an oil-in-water emulsion containing squalene, polysorbate 80, and sorbitan treolate, an adjuvant that activates the innate immune system [72].

This adjuvant is currently included in the adjuvanted trivalent (TIV) and quadrivalent (QIV) influenza vaccines marketed by Sequirus as Fluad [73].

MF59 can stimulate the production of inflammatory substances, increase the expression of molecules that help cells stick together, and activate genes involved in presenting antigens to the immune system [72].

Calabro et al. studied a vaccine adjuvant called MF59 and found that it effectively attracts various immune cells, including neutrophils, monocytes and dendritic cells, at the injection site [84]. These cells then carry the vaccine to the lymph nodes, where the immune response is initiated.

When compared to aluminium, MF59 injections resulted in a greater number of neutrophils being drawn to the injection site and a faster movement of the vaccine from the injection site to the lymph nodes [84].

MF59 activates local innate immune cells to secrete chemokines such as CCL4, CCL5, CCL25, and C-X-C motif ligand 8 (CXCL8). These chemokines initiate an adaptive immune response by promoting leukocyte recruitment, antigen uptake, and migration to lymph nodes [32].

The clinical manifestations of MF59-induced ASIA are nonspecific, including fever, arthralgia, myalgia, fatigue, chronic pain, depression, and sleep disorders, all of which contribute to a poor quality of life. In a minority of cases, patients present with systemic conditions that fulfill the criteria for various autoimmune diseases such as rheumatoid arthritis, systemic lupus erythematosus, systemic sclerosis, and overlap syndrome [32].

As described by Kuroda et al., adjuvant oils, such as pristane, incomplete Freund’s adjuvant (IFA) and MF59 may induce lupus-related autoantibodies in non-autoimmune mice, as well as pro-inflammatory cytokines, such as IL-6, IL-12, and TNF-α [85].

Particularly relevant are the effects recorded after the administration of the bivalent HPV vaccine.

In the preclinical randomized trial, 2881 women who received the bivalent HPV vaccine reported a higher number of deaths during the four-year follow-up compared to the 2871 women who received an “aluminum placebo”. A post hoc unblinded investigator opinion incorrectly dismissed the findings, concluding that none of the deaths were due to the vaccination [86]. A subsequent study by Arbyn et al. determined that there was no discernible pattern in the causes or timing of the deaths [87].

Dates from various locations, situations, and periods have shown a relationship between HPV vaccination and the onset of neuropathic pain and dysautonomia [32]. A possible explanation is that the dorsal root ganglia are sites of nervous system where various substances, including vaccines, can trigger these neurological conditions [88]. In a study with mice, Gilbert et al. showed that following vaccination, sensory neurons in the dorsal root ganglion were able to capture and retain antigen-specific antibodies released from antibody-secreting plasma cells [89].

It has also been associated with Asian endocrine disorders after HPV vaccination. Colafrancesco et al. documented three cases of primary ovarian failure (POF) following quadrivalent HPV vaccination. All cases met ASIA criteria and developed gastrointestinal symptoms, pain in the injected arm, arthralgia, depression, and headache followed by amenorrhea. Hormonal screening showed high levels of follicle-stimulating hormone (FSH) and luteinizing hormone (LH) and low levels of estradiol [90].

Another case of POF subsequent HPV vaccination was described by Little and Ward [42]. The patient experienced irregular menstruation after receiving the vaccine, which progressed to oligomenorrhea. The hormonal profile showed high FSH and LH levels and low estradiol and anti-müllerian hormone (AMH) levels [91].

Furthermore, rheumatological manifestations were described. For example, Watad et al. reported Henoch–Schönlein purpura following HPV vaccination [20].

Various reports describe ASIA-like symptoms in individuals who had been vaccinated against SARS-CoV-2 [92]. Several case reports describe the onset of endocrine diseases after vaccination. For instance, Pujol et al. reported three cases of thyroid issues following the first dose of the Pfizer/BioNTech COVID-19 vaccine [93].

Khan et al. reported the case of a woman who experienced fever, palpitations, and painful swelling on the left side of her neck four days after receiving the second dose of the Pfizer/BioNTech COVID-19 vaccine [94]. The patient also complained of right-sided neck pain and swelling after the first dose. Thyroid function tests showed hyperthyroidism and elevated inflammatory markers. Scintigraphy led to the diagnosis of subacute thyroiditis.

Siolos et al. also reported a case of thyroiditis after administration of the AstraZeneca vaccine [95].

However, the spectrum of ASIA manifestations following COVID-19 vaccination is very broad. For example, Abdelmaksound et al. reported 38 cases of vasculitis following COVID-19 vaccination [96]. Approximately half of them had been vaccinated with the mRNA vaccine. The most frequently reported subtypes of vasculitis are IgA and Leukocytoclastic vasculitis. Interestingly, mRNA COVID-19 vaccines are associated with developed ANCA-associated vasculitis [97].

It has been suggested that COVID-19 vaccination may induce age-related B cells (ABC cells) and trigger autoimmunity [98]. ABC cells, also called double negative B cells because they do not express the memory markers immunoglobulin D and CD27, are involved in increased autoantibody production; TLR-7 increases ABC cell activity and both TLR-7 and TLR-9 increase interferon I production. mRNA and DNA vaccines stimulate TLR-7 and TLR-9. As a result, vaccine-induced stimulation of ABC cells leads to the production of autoantibodies that may play a role in the development of post-vaccine autoimmune syndrome.

In their review of the literature, Chen et al. evaluated the autoimmune syndromes following COVID-19 vaccination; Guillain-Barre syndrome, vaccine-induced immune thrombotic thrombocytopenia, and rheumatic diseases were predominant [99]. The potential pathogenetic mechanisms are molecular mimicry, autoantibody production, and vaccine adjuvants. Among the last, it has been suggested an ‘adjuvant’ role of polyethylene glycol and polysorbate in stimulating the immune response [100].

HBV immunization has proved to cause various clinical manifestations, including neurological, musculoskeletal, muco-cutaneous, gastrointestinal, psychiatric, systemic and local reactions [101].

Interestingly, in a retrospective analysis conducted by Zafir et al., about 80% of 93 patients with immune-related illnesses following HBV vaccination showed elevated serum autoantibody titers [102].

Moreover, the administration of recombinant HBV vaccine has been linked to the development of systemic lupus erythematosus and acute disseminated encephalomyelitis, and the increase of anti-phospholipid antibodies [14].

However, much remains to be discovered about the autoimmune consequences of vaccination. In this context, the potential role of other less common human vaccines as triggers of immune-mediated disorders will be matter of future research.

**Table 2 vaccines-12-01183-t002:** The table below summarizes the different adjuvants, the vaccines they are associated with and their mechanisms of action.

Adjuvant	Vaccine	Mechanism of Action	References
**Aluminum**	Tetanus, influenza, pneumococcus, and hepatitis A and B	Triggering of NLRP3 inflammasome, enhancer of Th1 and Th2 immunity	[32,77]
**Polyoxyethylene sorbitan monooleate and sorbitan trioleate (MF59)**	Trivalent (TIV) and quadrivalent (QIV) influenza vaccines	Activation of neutrophils, monocytes and dendritic cells. Secretion of chemokines such as CCL4, CCL5, CCL25, and CXCL8	[32,84]
**HPV vaccination**		Sensory neurons in the dorsal root ganglion were able to capture and retain antigen-specific antibodies released from antibody-secreting plasma cells. Endocrine diseases.	[89,90]
**COVID-19 vaccination**		Induce age-related B cells (ABC cells) and trigger autoimmunity; stimulate TLR-7 and TLR-9	[98,99]

### 5.2. Miscellanea

Silicone is a synthetic polymer that can cause, after its injection, autoimmune reactions, but also local or systemic manifestations, such as siliconosis, calcinosis cutis with hypercalcemia, and chronic kidney disease (CKD). In literature, it has been reported a case of a young man developing ASIA, calcinosis cutis, and CKD after a sex-change surgical operation [103].

Silicone breast implants are recognized as a highly immunogenic adjuvant device; the most common diseases among SBI users are UCTD, Sjogren’s syndrome, autoimmune thyroiditis, systemic sclerosis, and adult-onset Still’s disease (AOSD) [104].

In vitro, it has been observed that peripheral blood mononuclear cells (PBMC) deriving from individuals undergoing inflammatory reactions after silicone injections produce higher levels of TNF-α and IL-6 [105]. Inside the body, silicones are oxidized to silica, activating macrophages and leading to the production of cytokines and free radicals [106].

Spite et al. have proposed that a local immune response, involving the suppression of Tregs and the activation of Th1/Th17 cells, may contribute to a pro-inflammatory environment and cytokine release. This inflammatory cascade could be a key factor in the progress of the classic symptoms of breast implants [107]. Additionally, silicone implants can lead to granulomatous disease, promoting the formation of capsules and fibrosis [108].

In women exposed to silicone from breast implants, Raynaud’s phenomenon was the most common reported problem, followed by sicca symptoms. Nailfold capillaroscopy tests suggested a possible link to scleroderma in some cases [109].

In addition, the study found that women with silicone breast implants often had elevated acute phase reactants, positive antinuclear antibodies, and antibodies associated with Sjögren’s syndrome. Several women met criteria for Sjögren’s syndrome or limited systemic sclerosis with symptoms, such as dry eyes/mouth, Raynaud’s phenomenon, and skin changes [109].

Cuellar et al. found an unusual ANA positivity (almost 58%) in individuals with silicone breast implants, using a HEp-2 cell line through an indirect immunofluorescence technique [110].

Plavsic et al. reported the development of ASIA in a woman with Hashimoto thyroiditis and familial autoimmunity who previously had a cosmetic silicone breast implantation [111].

Maina et al. reported three cases of AOSD associated with breast implantation [112]. In one case, the patient refused to remove the implant and only received medical treatment, thus experiencing numerous flares, as well as complications from glucocorticoid therapy. The other two patients had a complete resolution of symptoms through the combination of immunosuppressive therapy and silicone breast implant removal [112].

Based on current knowledge, breast silicone implants should not be indicated in individuals with autoimmune disorders, a previous autoimmune reaction to an adjuvant, and a genetic predisposition [113].

However, as highlighted by some recent meta-analyses, the results of the research about this topic are frequently contradictory or inconclusive, due to the small-sample size, the short follow-up of participants, the absent adjustment for confounders, and the rare distinction between silicone implants and other breast implants in these studies [114].

Hyaluronic acid (HA), commonly employed in aesthetic procedures, can be responsible for late immune-mediated reactions fulfilling the ASIA criteria. For example, as reported by Alijotas-Reig et al., among 25 patients with delayed immune reactions following HA injections, 5 patients developed clinical symptoms consistent with ASIA syndrome [115]. Moreover, Watad et al. examined 500 cases and discovered that HA was responsible for the syndrome in 29.2% [20].

Other articles have reported ASIA in susceptible individuals after various surgical procedures, including testicular implants, rhinoplasty, polypropylene mesh for hernia repair and pelvic floor augmentation, tension-free vaginal tape for stress urinary incontinence, prosthetic materials for joint replacement surgery, and metal implants used in orthopedic surgery have been reported to occur [116].

In a case report, Vaz et al. suggested that ‘metallosis’ could be a novel form of ASIA, characterized by the accumulation of heavy metal debris (e.g., cobalt and chromium) in soft tissues, caused by the presence of metal-on-metal prosthetic devices in the human body. This condition may be accompanied by both local and systemic symptoms, including chronic fatigue and neurological impairment [117]. Schiff et al. described two clinical cases of ASIA developed after a forearm fracture fixation surgery. In particular, the patients developed respectively an undifferentiated connective tissue disease and an overlap syndrome, and they had a clinical improvement after the surgical extraction of the metal implants [118].

Currently, the factors that determine a patient’s susceptibility to ASIA after medical device implantation are not fully understood. Various hypotheses have been proposed.

Patients with a history of allergies develop ASIA more frequently following implantation. Additionally, pre-existing autoimmune diseases or a family history of autoimmune conditions have a greater risk [43].

It has been observed that after implantation of the biomaterials, host proteins are absorbed and form a layer that facilitates the mobilization of macrophages [119].

Mast cells are also involved in this process through histamine production, which sensitizes nociceptors, and transient receptor potential vanilloid 1 (TRPV1) channels, causing pain at the implant site [43].

## 6. Treatments and Future Target

ASIA, including the concept of the exposome, should be addressed with a holistic approach.

The exposome refers to the totality of the environmental exposures to which an individual is exposed throughout the lifetime [120].

The original focus was on using objective methods to study how the environment affects human health [121]. This concept has been explored by researchers in a variety of fields, but the goal remains to quantify the combined effects of various exposures on human health outcomes [122].

The exposome concept seeks to complement the genome by considering how environmental factors interact with our genetic makeup to influence disease risk and potential treatment targets [123].

The evolving exposome has contributed to an increased risk of major diseases associated with immune dysfunction. Therefore, an integrated approach combining exposomic and genomic data would be valuable in improving the understanding of the etiology of ASIA, allowing for targeted treatments that address the environmental triggers of this condition.

Moreover, a detailed medical history should be conducted to investigate pre-existing autoimmune diseases or a family history of autoimmune conditions, as these factors also increase the risk of developing symptoms. Furthermore, it is very important to consider the coexistence of immunogenetic factors (e.g., HLA) and environmental factors such as smoking and obesity in the development of ASIA.

A better understanding of ASIA should lead to the development of safer vaccine adjuvants.

Systemic corticosteroids are the mainstay of treatment for acute and delayed immune-mediated adverse reaction. Moderate to high doses of prednisone (0.5–1 mg/kg/day) have consistently shown efficacy, with no refractory cases reported to date [2].

Alijotas-Reig et al. conducted a study using peripheral blood mononuclear cells (PBMCs) from individuals with granulomatous and autoimmune lesions associated with silicone, hyaluronic acid, and acrylamide [124].

Their research found that these substances can inhibit several proinflammatory cytokines, including TNF-α, IFN-γ, IL-1β, IL-2, and IL-6.

Indeed, Bindoli et al. reported four cases of hyperinflammatory symptoms associated with the ASIA spectrum following the administration of anti-SARS-CoV-2 mRNA vaccines. These cases were successfully treated with anakinra, an IL-1 receptor antagonist [3].

The rationale for using intravenous anakinra to treat the hyperinflammatory state is based on the results seen during the SARS-CoV-2 pandemic [125].

Concerning certain phenotypes resembling Still’s disease, the monoclonal anti-IL6-R antibody, such as Tocilizumab, could be used, inhibiting the signaling in the STAT3 cascade [126].

Finally, molecules that act as JAK inhibitors could represent a therapeutic strategy. For example, Wu et al. conducted studies in rats with adjuvant-induced arthritis using the JAK inhibitor SHR0302 and showed that SHR0302 can inhibit T and B cell proliferation [4]. SHR0302 also inhibits cytokines such as TNF-α, IL-1β, and IL-17, lowers IgG1 and IgG2 antibody levels, and suppresses the percentage of Th17 cells and total B cells [4].

Furthermore, finding new, safe adjuvants and tailoring them to patient profiles remains critical.

One area of focus is nanoformulations.

These advanced adjuvants have unique properties that can be tailored to enhance the immune response to vaccines [127]. The shift from traditional alum-based adjuvants to nanoformulations highlights the dynamic progress and potential of vaccine research. Innovations in adjuvant development, particularly with nanoformulations, represent a promising advancement toward improving both the efficacy and safety of vaccines.

Nanoparticles can optimize vaccine-induced immune responses by increasing antigen delivery and uptake by immune cells [128]. Polymeric nanoparticles can shield antigens and release them gradually, resulting in a more robust immune response. In addition, these nanoparticles can specifically target antigen-presenting cells to induce a stronger immune response [129].

The use of nanoparticles for vaccine delivery is an evolving field. Rigorous studies are needed to determine the validity of reported findings and their applicability to clinical settings. Although the properties of nanoparticles influence biological interactions, their precise effects remain unclear [130]. Rapid clearance of nanoparticles may limit their therapeutic potential. The development of methods to improve the reproducibility and persistence of nanoparticles in the body is essential to enhance cellular immunity [131].

## 7. Conclusions

Vaccines remain critical in the fight against infectious diseases by preventing transmission and reducing associated morbidity and mortality. However, it is important to consider the potential for vaccine-related adverse events, particularly autoimmune complications, especially in individuals with a genetic predisposition. Addressing these concerns would undoubtedly support prevention, early diagnosis and treatment efforts. In addition, understanding the risks and mechanisms underlying ASIA is key to developing vaccines with safer side effect profiles.

However, the criteria for ASIA are overly broad, leading to a lack of precision and making them difficult to apply and interpret in clinical practice. ASIA appears to include not only all patients with autoimmune diseases, but also a significant proportion of the general population with vague symptoms [132].

It is time for international Societies to assemble an independent panel of experts to objectively evaluate the diagnostic criteria for ASIA and the related supporting evidences.

Further researches are needed to elucidate the underlying pathophysiological mechanisms and to develop standardized management protocols for this complex condition.

Moreover, because this complex syndrome is relatively recently reported (2011), studies on larger, homogeneous populations are needed to clarify the pathogenesis and to establish unambiguous treatment protocols.

## Figures and Tables

**Figure 1 vaccines-12-01183-f001:**
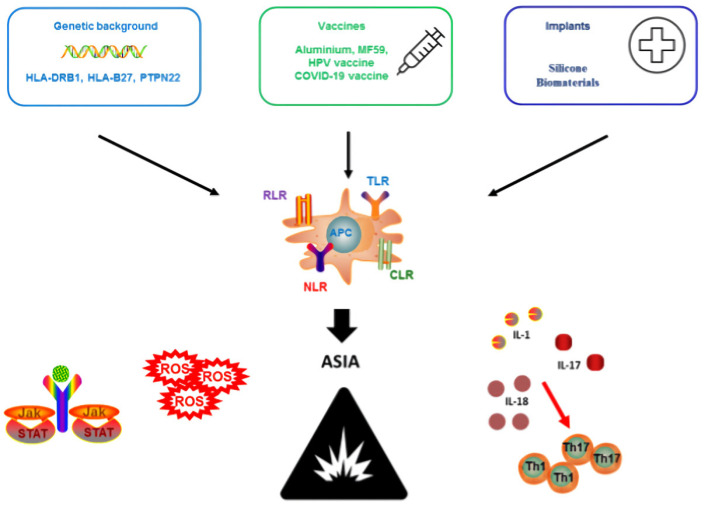
ASIA is a complex syndrome resulting from the interaction of genetic predisposition and environmental factors with adjuvants, through modulation of receptors, such as TLR, NLR and CLR, triggering aberrant immune responses and promoting the development of autoantibodies. Abbreviations: HLA, human leukocyte antigens; PTPN22, protein tyrosine phosphatase non-receptor type 22; HPV, human papillomavirus; RLR, retinoic acid-inducible gene I-like receptor; TLR, Toll-like receptor; NLR, NOD-like receptor; CLR, C-type lectin receptor; JAK/STAT, Janus kinase-transcriptional signal transducer activator; ROS, reactive oxygen species; IL, interleukin.

**Table 1 vaccines-12-01183-t001:** Criteria for the diagnosis of ASIA.

Major Criteria
Exposure to an external stimulus (infection, vaccine, silicone, adjuvant) before clinical manifestations
Presence of ‘typical’ clinical manifestations
Myalgia, myositis, or muscle weakness
Arthralgia and/or arthritis
Chronic fatigue, sleep disturbances
Neurological manifestations
Cognitive impairment, memory loss
Pyrexia, dry mouth
Improvement after removal of causing agent
Typical biopsy of involved organs
**Minor Criteria**
Autoantibodies or antibodies directed to the suspected adjuvant
Other clinical manifestations (i.e., irritable bowel syndrome)
Specific HLA (i.e., HLA DRB1, HLA DQB1)
Development of an autoimmune disease

Abbreviations: human leukocyte antigens (HLA).

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
