# Peer review of "ASIA Syndrome: State-of-the-Art and Future Perspectives"

_vaccines, 2024, doi:10.3390/vaccines12101183_

Round 1
Reviewer 1 Report
Comments and Suggestions for Authors
General Comments
- Introduction Enhancement: The introduction should provide a clearer overview of the review's scope and purpose. Currently, it jumps into details about ASIA without outlining the broader context or significance. I suggest starting with a definition of ASIA, its importance, and the key questions or topics to be addressed. Highlighting the review's relevance to vaccine development and public health policies, especially in light of global immunization strategies, would strengthen the introduction.
- Objective Clarification: The aim of the review should be articulated more clearly, with a stronger focus on the novel insights it provides. Please emphasize how your work adds new perspectives, particularly regarding new treatments, adjuvant innovations, and the impact of the COVID-19 pandemic.
- Reorganization of Subsections: The subsections within each main section (e.g., adjuvants, vaccines, treatment) could be more logically organized. A coherent structure will help readers better understand the relationships among the topics.
- Broader Epidemiological Overview: To strengthen the paper, consider providing a broader epidemiological overview of ASIA, addressing its global prevalence and diagnostic challenges. Discuss limitations of current research, emphasizing the need for larger, controlled studies and improved diagnostic criteria.
- Consistency in Formatting: Implement a consistent numbering system for all sections of the review. Currently, only sections from 4.2 are numbered, which may confuse readers. Adding numbers to earlier sections, including the introduction, would improve clarity and organization.
Specific Comments
- Clarification in the Introduction: While the conditions (GWS, MMF, SBS) are explained, the connection between each condition and the ASIA framework should be more explicit. It’s important to highlight how adjuvants contribute to these conditions within the ASIA context.
- Improvement of Sentences:
- In lines 63-65, revise: "We considered original and review articles, written in English since 2011, and published only in peer-reviewed journals" to "We considered only English-language original and review articles published in peer-reviewed journals since 2011."
- In the Epidemiology section, clarify the relevance of Hennekens et al.'s study by revising it to: "Hennekens et al. conducted a retrospective cohort study comparing 10,830 women with silicone breast implants to those without, assessing the risk of connective tissue diseases linked to ASIA."
- In the Epidemiology section, revise the sentence on diagnosis: "According to Shoenfeld et al., the diagnosis of ASIA requires two major criteria, or one major and two minor criteria, as outlined in Table 1."
- Pathophysiology Section: Provide clearer explanations of how genetic factors (HLA, PTPN22) interact with adjuvant exposure to trigger autoimmune responses in ASIA. Emphasize the relevance of animal models to human ASIA and the link between adjuvants like aluminum and immune processes (e.g., NLRP3 inflammasome activation).
- Figure Legend Improvement: Enhance the legend of Figure 1 by focusing on what the figure represents. For example: "Figure 1 illustrates how exposure to adjuvants triggers genetic and immunological processes that lead to autoimmune diseases in genetically susceptible individuals."
- Discussion on Adjuvants and ASIA: Clarify how specific adjuvants like aluminum and MF59 trigger autoimmune responses linked to ASIA. Expand on immune reactions from silicone and other implants and detail patient susceptibility factors like allergies and autoimmune history.
- Expand on the Exposome Concept: The mention of the exposome in the “Treatments and future targets” section is a good start, but expand it to provide practical insights. Explore how the exposome applies to the holistic management of ASIA.
- Treatment Options Clarification: Provide clearer explanations of how treatments like IL-1 and IL-17 inhibitors specifically address ASIA.
- Future Research Directions: Expand the "future targets" section to include ongoing research or novel adjuvant technologies. Reinforce the need for further research, particularly regarding new adjuvants and personalized treatment approaches.
- Stronger Conclusion: Add a more comprehensive conclusion section that summarizes key findings and future directions, emphasizing gaps in current research and areas needing attention.
- Visual Overview Addition: Consider including a table or flowchart summarizing different adjuvants, their mechanisms, and associated vaccines to enhance comprehension.
References
- Formatting and Completeness: Review the following references for formatting, completeness, and relevance:
- Reference 10: The format "S2444440524000967" does not clearly indicate volume or issue; please specify these details.
- Reference 4: Ensure consistent formatting (italics, capitalization) for the journal name and article title.
- Reference 29: For online sources, reformat this reference to start with "PubChem" and ensure a consistent style.
- Reference 33: The authors' names are not completed and seem incorrectly formatted.
- General Relevance: Evaluate the relevance of all references, particularly those that focus on general autoimmune diseases without a direct link to ASIA syndrome, such as:
- Reference 18: Mosca, M.; Tani, C.; Talarico, R.; Bombardieri, S.
- Reference 16: Schaefer, C.J.; Wooley, P.H.
- Reference 17: Zandman-Goddard, G.; Blank, M.; Ehrenfeld, M.; Gilburd, B.; Peter, J.; Shoenfeld, Y.
- Missing Reference: The reference "Vaccines and Autoimmunity - From Side Effects to ASIA Syndrome" by Seida et al. (2023) is missing but is highly relevant to your review. Including this would strengthen your literature foundation.
- Self-Citation Review: The reference by Di Renzo et al. () appears to be a self-citation. The title does not seem directly relevant to ASIA syndrome. Evaluate its contribution to your review's focus.
Comments on the Quality of English Language
The manuscript presents valuable content, but there are areas where clarity could be improved due to some awkward phrasing and minor grammatical errors. Moderate editing is recommended to enhance clarity and ensure that the key points are effectively communicated.
Author Response
Dear Editor of Vaccines
First, my coauthors and I would like to thank you sincerely for this opportunity to cooperate. We profoundly thank the reviewers for the comments and useful suggestions to improve the paper. We thank You for your constructive critique and hope the review process has improved the manuscript. If additional changes are warranted, we will make them. 
We hope that this revised version of our manuscript may now be found suitable for publication. 
This is a point-by-point list of changes made in the paper:
REVIEWER 1
General Comments
- Introduction Enhancement: The introduction should provide a clearer overview of the review's scope and purpose. Currently, it jumps into details about ASIA without outlining the broader context or significance. I suggest starting with a definition of ASIA, its importance, and the key questions or topics to be addressed. Highlighting the review's relevance to vaccine development and public health policies, especially in light of global immunization strategies, would strengthen the introduction.
We modified the introduction, as suggested;
- Objective Clarification: The aim of the review should be articulated more clearly, with a stronger focus on the novel insights it provides. Please emphasize how your work adds new perspectives, particularly regarding new treatments, adjuvant innovations, and the impact of the COVID-19 pandemic.
We emphasized the goal of the work, as required;
- Reorganization of Subsections: The subsections within each main section (e.g., adjuvants, vaccines, treatment) could be more logically organized. A coherent structure will help readers better understand the relationships among the topics.
We have changed the division into paragraphs hoping to have improved the readability of the paper;
- Broader Epidemiological Overview: To strengthen the paper, consider providing a broader epidemiological overview of ASIA, addressing its global prevalence and diagnostic challenges. Discuss limitations of current research, emphasizing the need for larger, controlled studies and improved diagnostic criteria.
We have expanded the section on epidemiology, as requested;
- Consistency in Formatting: Implement a consistent numbering system for all sections of the review. Currently, only sections from 4.2 are numbered, which may confuse readers. Adding numbers to earlier sections, including the introduction, would improve clarity and organization.
We numbered all the sections of the paper.
Specific Comments
- Clarification in the Introduction: While the conditions (GWS, MMF, SBS) are explained, the connection between each condition and the ASIA framework should be more explicit. It’s important to highlight how adjuvants contribute to these conditions within the ASIA context.
We explained how the pathologies mentioned are related to ASIA, as required;
- Improvement of Sentences:
- In lines 63-65, revise: "We considered original and review articles, written in English since 2011, and published only in peer-reviewed journals" to "We considered only English-language original and review articles published in peer-reviewed journals since 2011."
- In the Epidemiology section, clarify the relevance of Hennekens et al.'s study by revising it to: "Hennekens et al. conducted a retrospective cohort study comparing 10,830 women with silicone breast implants to those without, assessing the risk of connective tissue diseases linked to ASIA."
- In the Epidemiology section, revise the sentence on diagnosis: "According to Shoenfeld et al., the diagnosis of ASIA requires two major criteria, or one major and two minor criteria, as outlined in Table 1."
We modified the highlighted sentences, as suggested;
- Pathophysiology Section: Provide clearer explanations of how genetic factors (HLA, PTPN22) interact with adjuvant exposure to trigger autoimmune responses in ASIA. Emphasize the relevance of animal models to human ASIA and the link between adjuvants like aluminum and immune processes (e.g., NLRP3 inflammasome activation).
We have explained how genetic factors are predisposing and added additional references on the effects of aluminum on the immune system, as requested;
- Figure Legend Improvement: Enhance the legend of Figure 1 by focusing on what the figure represents. For example: "Figure 1 illustrates how exposure to adjuvants triggers genetic and immunological processes that lead to autoimmune diseases in genetically susceptible individuals."
We have modified the figure legend, as requested;
- Discussion on Adjuvants and ASIA: Clarify how specific adjuvants like aluminum and MF59 trigger autoimmune responses linked to ASIA. Expand on immune reactions from silicone and other implants and detail patient susceptibility factors like allergies and autoimmune history.
We have expanded the role of these adjuvants, as required;
- Expand on the Exposome Concept: The mention of the exposome in the “Treatments and future targets” section is a good start but expand it to provide practical insights. Explore how the exposome applies to the holistic management of ASIA.
We expanded and explained how the concept of exposome can be employed in the approach to ASIA;
- Treatment Options Clarification: Provide clearer explanations of how treatments like IL-1 and IL-17 inhibitors specifically address ASIA.
We explained how anti IL-1 drugs can be used in the treatment of ASIA; due to lack of solid evidence, we removed references for IL-17;
- Future Research Directions: Expand the "future targets" section to include ongoing research or novel adjuvant technologies. Reinforce the need for further research, particularly regarding new adjuvants and personalized treatment approaches.
We have expanded the section and included references for new technologies regarding adjuvants;
- Stronger Conclusion: Add a more comprehensive conclusion section that summarizes key findings and future directions, emphasizing gaps in current research and areas needing attention.
We have created a new paragraph for ‘Conclusions’;
- Visual Overview Addition: Consider including a table or flowchart summarizing different adjuvants, their mechanisms, and associated vaccines to enhance comprehension.
We have added a table on adjuvants and vaccines, as requested.
References
- Formatting and Completeness: Review the following references for formatting, completeness, and relevance:
- Reference 10: The format "S2444440524000967" does not clearly indicate volume or issue; please specify these details.
- Reference 4: Ensure consistent formatting (italics, capitalization) for the journal name and article title.
- Reference 29: For online sources, reformat this reference to start with "PubChem" and ensure a consistent style.
- Reference 33: The authors' names are not completed and seem incorrectly formatted.
We formatted the indicated references, as requested;
- General Relevance: Evaluate the relevance of all references, particularly those that focus on general autoimmune diseases without a direct link to ASIA syndrome, such as:
- Reference 18: Mosca, M.; Tani, C.; Talarico, R.; Bombardieri, S.
- Reference 16: Schaefer, C.J.; Wooley, P.H.
- Reference 17: Zandman-Goddard, G.; Blank, M.; Ehrenfeld, M.; Gilburd, B.; Peter, J.; Shoenfeld, Y.
We evaluated these references, that we find useful in the discussion of the paper;
- Missing Reference: The reference "Vaccines and Autoimmunity - From Side Effects to ASIA Syndrome" by Seida et al. (2023) is missing but is highly relevant to your review. Including this would strengthen your literature foundation.
We included this reference.
- Self-Citation Review: The reference by Di Renzo et al. () appears to be a self-citation. The title does not seem directly relevant to ASIA syndrome. Evaluate its contribution to your review's focus.
We have removed this reference.
Sincerely
Rossella Cianci
Reviewer 2 Report
Comments and Suggestions for Authors
The manuscript is a review on adjuvant-induced autoimmune/inflammatory syndrome (ASIA), a condition also known as Shoenfeld Syndrome that was first described in 2011 by Yehuda Shoelfeld in genetically predisposed individuals exposed to compounds such as silicone, drugs, metals, and vaccine adjuvants. This work emphasizes the role of adjuvants in vaccines, especially after COVID-19 vaccination, and addresses new treatment approaches.
Strengths:
- Timeliness and Relevance: The topic of Autoimmune/Inflammatory Syndrome Induced by Adjuvants (ASIA) is highly relevant, especially with the increased attention on vaccine safety and immunological reactions in the wake of mass vaccination campaigns for COVID-19. The manuscript provides a thorough literature review on this topic.
Weaknesses:
- The content is a summary of existing literature. While informative, the authors should aim to provide more original insights, even in a review, to make it more impactful. The methodology for the literature review is poorly detailed, lacking clear inclusion/exclusion criteria for the studies, making it hard to assess the review's reliability and comprehensiveness.
-The author's analysis relies on specific studies or cases, such as those related to silicone implants and HPV vaccines, without discussing the broader context of other potential triggers in sufficient detail. This may give readers the impression that these are the predominant causes, whereas ASIA can have multiple triggers across various populations.
- The manuscript references studies of varying methodological quality without critically evaluating them, treating self-reported and small-sample studies with the same weight as more rigorous research. A more critical discussion of study limitations, especially on controversial topics like the link between silicone implants and ASIA, is needed.
- The discussion of ASIA's pathophysiology is lacking in depth, particularly in how specific adjuvants, like aluminum salts, interact with the immune system. While general mechanisms such as inflammasome activation are mentioned, a more detailed molecular explanation is needed.
- Some sections are repetitive, especially regarding the pathophysiology and the role of adjuvants. For instance, the role of aluminum-based adjuvants is discussed multiple times without adding new insights.
- Despite the current interest in COVID-19 vaccination I think the authors disproportionately focus on COVID-19 vaccines compared to other vaccines and adjuvants. Since various adjuvants can trigger ASIA, this emphasis may create an unbalanced perspective of the syndrome.
In summary, the manuscript offers a useful overview of ASIA but needs more originality, a critical literature assessment, and a deeper exploration of molecular mechanisms. Structural, content, and balance revisions could make it a comprehensive review of the topic
Comments on the Quality of English LanguageMinor editing of English language required.
Author Response
Dear Editor of Vaccines
First, my coauthors and I would like to thank you sincerely for this opportunity to cooperate. We profoundly thank the reviewers for the comments and useful suggestions to improve the paper. We thank You for your constructive critique and hope the review process has improved the manuscript. If additional changes are warranted, we will make them. 
We hope that this revised version of our manuscript may now be found suitable for publication. 
This is a point-by-point list of changes made in the paper:
REVIEWER 2
The manuscript is a review on adjuvant-induced autoimmune/inflammatory syndrome (ASIA), a condition also known as Shoenfeld Syndrome that was first described in 2011 by Yehuda Shoelfeld in genetically predisposed individuals exposed to compounds such as silicone, drugs, metals, and vaccine adjuvants. This work emphasizes the role of adjuvants in vaccines, especially after COVID-19 vaccination, and addresses new treatment approaches.
Strengths:
- Timeliness and Relevance: The topic of Autoimmune/Inflammatory Syndrome Induced by Adjuvants (ASIA) is highly relevant, especially with the increased attention on vaccine safety and immunological reactions in the wake of mass vaccination campaigns for COVID-19. The manuscript provides a thorough literature review on this topic.
Weaknesses:
- The content is a summary of existing literature. While informative, the authors should aim to provide more original insights, even in a review, to make it more impactful. The methodology for the literature review is poorly detailed, lacking clear inclusion/exclusion criteria for the studies, making it hard to assess the review's reliability and comprehensiveness.
We have better clarified the choice of articles conducted on numerically smaller populations.
-The author's analysis relies on specific studies or cases, such as those related to silicone implants and HPV vaccines, without discussing the broader context of other potential triggers in sufficient detail. This may give readers the impression that these are the predominant causes, whereas ASIA can have multiple triggers across various populations.
- We discussed the various adjuvants and vaccines involved in the development of ASIA.
- The manuscript references studies of varying methodological quality without critically evaluating them, treating self-reported and small-sample studies with the same weight as more rigorous research. A more critical discussion of study limitations, especially on controversial topics like the link between silicone implants and ASIA, is needed.
We have modified the text, as requested.
- The discussion of ASIA's pathophysiology is lacking in depth, particularly in how specific adjuvants, like aluminum salts, interact with the immune system. While general mechanisms such as inflammasome activation are mentioned, a more detailed molecular explanation is needed.
We have expanded the sections on pathophysiology, as requested.
- Some sections are repetitive, especially regarding the pathophysiology and the role of adjuvants. For instance, the role of aluminum-based adjuvants is discussed multiple times without adding new insights.
We eliminated redundancies, as requested.
- Despite the current interest in COVID-19 vaccination I think the authors disproportionately focus on COVID-19 vaccines compared to other vaccines and adjuvants. Since various adjuvants can trigger ASIA, this emphasis may create an unbalanced perspective of the syndrome.
We have expanded the sections, adding other adjuvants and vaccines, as requested.
In summary, the manuscript offers a useful overview of ASIA but needs more originality, a critical literature assessment, and a deeper exploration of molecular mechanisms. Structural, content, and balance revisions could make it a comprehensive review of the topic.
Sincerely
Rossella Cianci
Round 2
Reviewer 1 Report
Comments and Suggestions for Authors
I appreciate the authors' efforts in addressing the major modifications. The revisions have significantly improved the manuscript's clarity and organization. The connections to ASIA are now clearer, and the conclusion effectively summarizes the key points.
I encourage the authors to make minor edits for grammar and clarity where needed. Thank you for your hard work on this important topic.
Comments on the Quality of English LanguageWhile the overall clarity and readability are generally good, some phrases could still benefit from minor edits to enhance the overall flow and precision of the text. A final review for grammar and consistency is advisable to ensure that the language is polished and professional throughout.
Reviewer 2 Report
Comments and Suggestions for Authors
The manuscript has been thoroughly revised in accordance with the critiques and suggestions. All comments have been carefully addressed, and appropriate changes have been made to improve the work's clarity, structure, and overall quality. We believe that the revisions have strengthened the manuscript and I´m confident that it now meets the required standards for publication.
Comments on the Quality of English LanguageMinor editing of English language required.